# CAUSAL DISENTANGLED REPRESENTATION LEARNING WITH VAE AND CAUSAL FLOWS

## ABSTRACT

Disentangled representation learning aims to learn a low dimensional representation of data where each dimension corresponds to one underlying generative factor. Due to the causal relationships between generative factors in real-world situations, causal disentangled representation learning has received widespread attention. In this paper, we first propose a variant of autoregressive flows, called causal flows, which incorporate true causal structure of generative factors into the flows. Then, we design a new VAE model based on causal flows named *Causal Flows Variational Autoencoders (CauF-VAE)* to learn causally disentangled representations. We provide a theoretical analysis of the disentanglement identifiability of CauF-VAE by incorporating supervised information on the ground-truth factors. The performance of CauF-VAE is evaluated on both synthetic and real datasets, showing its capability of achieving causal disentanglement and performing intervention experiments. Moreover, CauF-VAE exhibits remarkable performance on downstream tasks and has the potential to learn true causal structure among factors.

## 1 INTRODUCTION

Representation learning aims to create data representations that simplify information extraction for constructing classifiers or predictors (Bengio et al., 2013). Disentangled representation learning is an important step towards better representation learning. It assumes that high dimensional data is generated by low dimensional, semantically meaningful factors, called ground-truth factors. Thus, disentangled representation learning refers to learning a representation where changes in one dimension are caused by only one factor of variation in the data (Locatello et al., 2019a). The common framework for obtaining disentangled representations is Variational Autoencoders (VAE) (Kingma & Welling, 2013).

Recently, many unsupervised methods of learning disentangled representations with VAE have been proposed, mainly by imposing independent constraints on the posterior or aggregate posterior of latent variables $\mathbf{z}$ through KL divergence (Higgins et al., 2017; Burgess et al., 2018; Kim & Mnih, 2018; Chen et al., 2018; Kumar et al., 2017; Kim et al., 2019; Dupont, 2018). Later, Locatello et al. (2019a) proposed that it is almost impossible to achieve disentanglement in an unsupervised manner without inductive bias. Therefore, some weakly supervised or supervised methods were proposed (Locatello et al., 2019b; Shen et al., 2022; Yang et al., 2021; Khemakhem et al., 2020a; Locatello et al., 2020). However, most of them assumed that the generative factors are independent of each other, while in the real world, the generative factors of interest are likely to have causal relationships with each other. At this point, the above models can't achieve disentanglement of causally related factors (Shen et al., 2022; Träuble et al., 2021). To address this, recent state-of-the-art methods have focused on causal disentangled representation learning (Shen et al., 2022; Yang et al., 2021; Suter et al., 2019; Reddy et al., 2022; An et al., 2023; Brehmer et al., 2022; Buchholz et al., 2023; Lippe et al., 2022; Seigal et al., 2022). However, many of these methods either rely on existing Structural Causal Models (SCMs) to introduce causal relationships or focus solely on modeling the causal structure of generative factors in specific scenarios.

Normalizing flows (Rezende & Mohamed, 2015)), particularly autoregressive flows (Huang et al., 2018), have improved the inference capability of VAEs (Kingma et al., 2016) by approximating the true posterior distribution more closely. Notably, autoregressive flows have shown promise in learning causal orders (Khemakhem et al., 2021). Inspired by this application, we develop causal flows that

can impose causal structure on the variables. In essence, the causal flows incorporate information about how ground-truth factors are causally related. We then integrate causal flows into the VAE to help learn the disentangled representations.

In this paper, we first introduce causal flows and then propose a causal disentanglement model, called CauF-VAE, which combines VAE with causal flows to learn causally disentangled representations. After encoding the input data by VAE's encoder and processing it through causal flows, we obtain the causally disentangled representations, which will be fed into the decoder for the reconstruction of the original image. Our model avoids imposing restrictions on the causal structure among generative factors, enabling a more accurate and widely applicable modeling of real-world scenarios. To theoretically guarantee causal disentanglement, we incorporate supervised information about the underlying factors.

Our main results are summarized as follows:

    1) We introduce causal flows, the improved autoregressive flows that integrate causal structure information of ground-truth factors.

    2) We propose a new VAE model, *Causal Flows Variational Autoencoders(CauF-VAE)*, which employs the causal flows to learn causally disentangled representations.

    3) We theoretically prove that CauF-VAE satisfies disentanglement identifiability[1].

    4) We empirically show that CauF-VAE can achieve causal disentanglement and perform intervention experiments on both synthetic and real datasets. Our model exhibits excellent performance in terms of sampling efficiency and distributional robustness in downstream tasks, and further experiments indicate its potential to learn true causal structure.

## 2 BACKGROUND

### 2.1 VARIATIONAL AUTOENCODERS

Let $\{\mathbf{x}_j\}_{j=1}^N$ denote i.i.d training data, $\mathbf{x} \in \mathbb{R}^n$ be the observed variables and $\mathbf{z} \in \mathbb{R}^d$ be the latent variables. The dataset $\mathcal{X}$ has an empirical data distribution denoted as $q_{\mathcal{X}}$. The *generative model* defined over $\mathbf{x}$ and $\mathbf{z}$ is $p_{\boldsymbol{\theta}}(\mathbf{x}, \mathbf{z}) = p(\mathbf{z})p_{\boldsymbol{\theta}}(\mathbf{x}|\mathbf{z})$, where $\boldsymbol{\theta}$ is the parameter of the *decoder*. Typically, $p(\mathbf{z}) = \mathcal{N}(\mathbf{0}, \mathbf{I})$, $p_{\boldsymbol{\theta}}(\mathbf{x}|\mathbf{z}) = \mathcal{N}(f_{\boldsymbol{\theta}}(\mathbf{z}), \sigma^2\mathbf{I})$, where $f_{\boldsymbol{\theta}}(\mathbf{z})$ is a neural network. Then, the marginal likelihood $p_{\boldsymbol{\theta}}(\mathbf{x}) = \int p_{\boldsymbol{\theta}}(\mathbf{x}, \mathbf{z})d\mathbf{z}$ is intractable to maximize. Therefore, VAE (Kingma & Welling, 2013) introduces a parametric *inference model* $q_{\boldsymbol{\phi}}(\mathbf{z}|\mathbf{x}) = \mathcal{N}(\mu_{\boldsymbol{\phi}}(\mathbf{x}), \text{diag}(\sigma_{\boldsymbol{\phi}}^2(\mathbf{x})))$, also called an *encoder* or a *recognition model*, to obtain the variational lower bound on the marginal log-likelihood, i.e., the Evidence Lower BOund (ELBO):

$$
\begin{aligned}
\text{ELBO}(\boldsymbol{\phi}, \boldsymbol{\theta}) &= \mathbb{E}_{q_{\mathcal{X}}}\left[\log p_{\boldsymbol{\theta}}(\mathbf{x}) - D_{\text{KL}}(q_{\boldsymbol{\phi}}(\mathbf{z}|\mathbf{x})\|p_{\boldsymbol{\theta}}(\mathbf{z}|\mathbf{x}))\right] \\
&= \mathbb{E}_{q_{\mathcal{X}}}\left[\mathbb{E}_{q_{\boldsymbol{\phi}}(\mathbf{z}|\mathbf{x})}(\log p_{\boldsymbol{\theta}}(\mathbf{x}, \mathbf{z}) - \log q_{\boldsymbol{\phi}}(\mathbf{z}|\mathbf{x}))\right] \\
&= \mathbb{E}_{q_{\mathcal{X}}}\left[\mathbb{E}_{q_{\boldsymbol{\phi}}(\mathbf{z}|\mathbf{x})}\log p_{\boldsymbol{\theta}}(\mathbf{x}|\mathbf{z}) - D_{\text{KL}}(q_{\boldsymbol{\phi}}(\mathbf{z}|\mathbf{x})\|p(\mathbf{z}))\right]
\end{aligned} \tag{1}
$$

As can be seen from equation (1), maximizing $\mathcal{L}(\mathbf{x}, \boldsymbol{\phi}, \boldsymbol{\theta})$ will simultaneously maximize $\log p_{\boldsymbol{\theta}}(\mathbf{x})$ and minimize $D_{\text{KL}}(q_{\boldsymbol{\phi}}(\mathbf{z}|\mathbf{x})\|p_{\boldsymbol{\theta}}(\mathbf{z}|\mathbf{x})) \geq 0$. Therefore, we wish $q_{\boldsymbol{\phi}}(\mathbf{z}|\mathbf{x})$ to be flexible enough to match the true posterior $p_{\boldsymbol{\theta}}(\mathbf{z}|\mathbf{x})$. At the same time, based on the third line of equation (1), which is often used as objective function of VAE, we require that $q_{\boldsymbol{\phi}}(\mathbf{z}|\mathbf{x})$ is efficiently computable, differentiable, and sampled from.

### 2.2 AUTOREGRESSIVE NORMALIZING FLOWS

*Normalizing flows* (Rezende & Mohamed, 2015) are effective solutions to the issues mentioned above. The flows construct flexible posterior distribution through expressing $q_{\boldsymbol{\phi}}(\mathbf{z}|\mathbf{x})$ as an expressive invertible and differentiable mapping $\boldsymbol{g}$ of a random variable with a relatively simple distribution, such as an isotropic Normal. Typically, $\boldsymbol{g}$ is obtained by composing a sequence of invertible and differentiable transformations $\boldsymbol{g}_1, \boldsymbol{g}_2, \ldots, \boldsymbol{g}_K$, i.e., $\boldsymbol{g} = \boldsymbol{g}_K \circ \cdots \circ \boldsymbol{g}_1, \boldsymbol{g}_k : \mathbb{R}^{d+n} \to \mathbb{R}^d, \forall k = 1 \ldots K$.

---

[1]We adopt the definition of disentanglement and model's identifiability in Shen et al. (2022), which differs from that in Khemakhem et al. (2020a) in terms of goals and assumptions.

If we define the initial random variable (the output of encoder) as $\mathbf{z}_0$ and the final output random variable as $\mathbf{z}_K$, then $\mathbf{z}_k = \boldsymbol{g}_k(\mathbf{z}_{k-1}, \mathbf{x}), \forall k$. In this case, we can use $\boldsymbol{g}$ to obtain the conditional probability density function of $\mathbf{z}_K$ by applying the general probability-transformation formula (Papamakarios et al., 2021):

$$q_\phi(\mathbf{z}_K|\mathbf{x}) = q_\phi(\mathbf{z}_0|\mathbf{x}) \left| \det J_{\boldsymbol{g}(\mathbf{z}_0, \mathbf{x})} \right| \tag{2}$$

where $\det J_{\boldsymbol{g}(\mathbf{z}_0, \mathbf{x})}$ is the Jacobian determinants of $\boldsymbol{g}$ with respect to $\mathbf{z}_0$.

*Autoregressive flows* are one of the most popular normalizing flows(Huang et al., 2018; Papamakarios et al., 2021; Kingma et al., 2016). By carefully designing the function $\boldsymbol{g}$, the Jacobian matrix in equation (2) becomes a lower triangular matrix. For illustration, we will only use a single-step flow with notation $\boldsymbol{g}$. Multi-layer flows are simply the composition of the function represented by a single-step flow, as mentioned earlier. And we will denote the input to the function $\boldsymbol{g}$ as $\mathbf{z}$ and its output as $\widetilde{\mathbf{z}}$. In the autoregressive flows, $\boldsymbol{g}$ has the following form:

$$\widetilde{\mathbf{z}} = \boldsymbol{g}(\mathbf{z}, \mathbf{x}) = \left[ g^1(\mathbf{z}^1; \boldsymbol{h}^1) \dots g^d(\mathbf{z}^d; \boldsymbol{h}^d) \right]^{\mathrm{T}} \quad \text{where} \quad \boldsymbol{h}^i = \boldsymbol{c}^i(\widetilde{\mathbf{z}}^{<i}, \mathbf{x}) \tag{3}$$

where $g^i$, an invertible function of input $\mathbf{z}^i$, is termed as a **transformer**. Here $\mathbf{z}^i$ stands for the i-th element of vector $\mathbf{z}$, and $\boldsymbol{c}^i$ is the i-th **conditioner**, a function of the first $i-1$ elements of $\widetilde{\mathbf{z}}$, which determines part of parameters of the transformer $g^i$. We use neural networks to fit $\boldsymbol{c}$.

**Performing causal inference tasks**  The ordering of variables in autoregressive flows can be explained using the Structural Equation Models (SEMs) (Khemakhem et al., 2021; Pearl, 2009a), due to the similarity between equation (3) and the SEMs. Moreover, we could efficiently learn the causal direction between two variables or pairs of multivariate variables from a dataset by using autoregressive flows according to Khemakhem et al. (2021).

## 3 CAUSAL FLOWS

Motivated by Khemakhem et al. (2021), we propose an extension to the autoregressive flows by incorporating an adjacency matrix $A$. The extended flows still involve functions with tractable Jacobian determinants.

In autoregressive flows, causal order is established among variables $\widetilde{\mathbf{z}}^1, \cdots, \widetilde{\mathbf{z}}^d$. Given the causal graph of the variables $\widetilde{\mathbf{z}}^1, \cdots, \widetilde{\mathbf{z}}^d$, let $A \in \mathbb{R}^{d \times d}$ denote its corresponding binary adjacency matrix, $A_{i,:}$ is the row vector of $A$ and $A_{i,j}$ is nonzero only if $\widetilde{\mathbf{z}}^j$ is the parent node of $\widetilde{\mathbf{z}}^i$, then $A$ corresponding to the causal order in autoregressive flows is a full lower-triangular matrix. The conditioner can be written in the form of $\boldsymbol{c}^i(\widetilde{\mathbf{z}} \circ A, \mathbf{x})$, where $\circ$ is the element-wise product. If we utilize prior knowledge about the true causal structure among variables, i.e., if a certain causal structure among variables is known, then $A$ is still a lower triangular matrix, but some of its entries are set to 0 according to the underlying causal graph. We can integrate such $A$ into the conditioner, which is also denoted as $\boldsymbol{c}^i(\widetilde{\mathbf{z}} \circ A, \mathbf{x})$. We will refer to it as the **causal conditioner** in the following.

We define autoregressive flows that use the causal conditioner as **Causal Flows**. The transformer can be any invertible function, and we focus on affine transformer, which is one of the simplest transformers. Therefore, causal flows $\boldsymbol{g}$ can be formulated as follows:

$$\widetilde{\mathbf{z}}^i = g^i(\mathbf{z}^i; \boldsymbol{h}^i) = \mathbf{z}^i \exp(s_i(\widetilde{\mathbf{z}} \circ A_{i,:}, \mathbf{x})) + t_i(\widetilde{\mathbf{z}} \circ A_{i,:}, \mathbf{x}) \tag{4}$$

where $\boldsymbol{s} = [s_1, \cdots, s_d]^{\mathrm{T}} \in \mathbb{R}^d$ and $\boldsymbol{t} = [t_1, \cdots, t_d]^{\mathrm{T}} \in \mathbb{R}^d$ are defined by the conditioner, i.e., $\boldsymbol{h}^i = \{s_i, t_i\}$, while $s_1$ and $t_1$ are constants.

Given that the derivative of the transformer with respect to $\mathbf{z}^i$ is $\exp(s_i(\widetilde{\mathbf{z}} \circ A_{i,:}, \mathbf{x}))$ and $A$ is lower-triangular, the log absolute Jacobian determinant is:

$$\log \left| \det J_{\boldsymbol{g}(\mathbf{z}, \mathbf{x})} \right| = \sum_{i=1}^d \log \exp(s_i(\widetilde{\mathbf{z}} \circ A_{i,:}, \mathbf{x})) = \sum_{i=1}^d s_i(\widetilde{\mathbf{z}} \circ A_{i,:}, \mathbf{x}) \tag{5}$$

Now, we are able to derive the log probability density function of $\widetilde{\mathbf{z}}$ using the following expression:

$$\log q_\phi(\widetilde{\mathbf{z}}|\mathbf{x}) = \log q_\phi(\mathbf{z}|\mathbf{x}) - \sum_{i=1}^d s_i(\widetilde{\mathbf{z}} \circ A_{i,:}, \mathbf{x}) \tag{6}$$

It is worth emphasizing that the computation of autoregressive flows in equation (3) needs to be performed sequentially, meaning that $\widetilde{\mathbf{z}}^{<i}$ must be calculated before $\widetilde{\mathbf{z}}^{i}$. Due to the sampling requirement in VAE, this approach may not be computationally efficient. However, in causal disentanglement applications of VAE, the number of factors of interest is often relatively small. Additionally, we've found that using a single layer of causal flows and lower-dimensional latent variables are enough to lead to better results, so the computational cost of the model is not significantly affected by sequential sampling.

## 4 CauF-VAE: Causal Flows for VAE Disentanglement

This section focuses on addressing the issue of causal disentanglement in VAE. The main approach is to introduce causal flows into VAE. Moreover, we will incorporate supervised information into the model to achieve the alignment between latent variables and the ground-truth factors, ultimately achieving causal disentanglement.

First, we introduce some notations following Shen et al. (2022). We denote $\boldsymbol{\xi} \in \mathbb{R}^m$ as the underlying ground-truth factors of interest for data $\mathbf{x}$, with distribution $p_{\boldsymbol{\xi}}$. For each underlying factor $\boldsymbol{\xi}^i$, we denote $\boldsymbol{y}^i$ as some continuous or discrete annotated observation satisfying $\boldsymbol{\xi}^i = \mathbb{E}(\boldsymbol{y}^i|\mathbf{x})$, where the superscript $i$ still denotes the i-th element of each vector. Let $\mathcal{D} = \{(\mathbf{x}_j, \boldsymbol{y}_j, \mathbf{u}_j)\}_{j=1}^N$ denotes a labeled dataset, where $\mathbf{u}_j \in \mathbb{R}^k$ is the additional observed variable. Depending on the context, the variable $\mathbf{u}$ can take on various meanings, such as serving as the time index in a time series, a class label, or another variable that is observed concurrently (Hyvarinen & Morioka, 2016). We get $\boldsymbol{\xi}^i = \mathbb{E}(\boldsymbol{y}^i|\mathbf{x}, \mathbf{u})$, where $i = 1, \cdots, m$. This is because if $\mathbf{u}$ is ground-truth factor $\boldsymbol{y}$, it is obviously true, otherwise, $\boldsymbol{\xi}^i = \mathbb{E}(\boldsymbol{y}^i|\mathbf{x}, \mathbf{u}) = \mathbb{E}(\boldsymbol{y}^i|\mathbf{x})$. We will view the encoder and flows as an unified stochastic transformation $E$, with the learned representation $\widetilde{\mathbf{z}}$ as its final output, i.e., $\widetilde{\mathbf{z}} = E(\mathbf{x}, \mathbf{u})$. Additionally, in the stochastic transformation $E(\mathbf{x}, \mathbf{u})$, we use $\bar{E}(\mathbf{x}, \mathbf{u})$ to denote its deterministic part, i.e., $\bar{E}(\mathbf{x}, \mathbf{u}) = \mathbb{E}(E(\mathbf{x}, \mathbf{u})|\mathbf{x}, \mathbf{u})$. Now, we adopt the definition of causal disentanglement as follows:

**Definition 1 (Disentangled representation )**  *Considering the underlying factor $\boldsymbol{\xi} \in \mathbb{R}^m$ of data $\mathbf{x}$, $E$ is said to learn a disentangled representation with respect to $\boldsymbol{\xi}$ if there exists a one-to-one function $r_i$ such that $\bar{E}(\mathbf{x}, \mathbf{u})^i = r_i(\boldsymbol{\xi}^i), \forall i = 1, \cdots, m$.*

The purpose of this definition is to guarantee some degree of alignment between the latent variable $E(\mathbf{x})$ and the underlying factor $\boldsymbol{\xi}$ in the model. In our approach, we will also supervise each latent variable with label for each underlying factor, thus establishing such component-wise relationship between them.

### 4.1 CauF-VAE

We now proceed to present the full probabilistic formulation of CauF-VAE. The model's structure is depicted in Figure 1. The conditional generative model is defined as follows:

$$
\begin{array}{rcl}
p_{\boldsymbol{\theta}}(\mathbf{x}, \widetilde{\mathbf{z}}|\mathbf{u}) & = & p_{\mathbf{f}}(\mathbf{x}|\widetilde{\mathbf{z}}, \mathbf{u})p_{\mathbf{T}, \boldsymbol{\lambda}}(\widetilde{\mathbf{z}}|\mathbf{u}) \quad (7) \\
p_{\mathbf{f}}(\mathbf{x}|\widetilde{\mathbf{z}}, \mathbf{u}) & = & p_{\mathbf{f}}(\mathbf{x}|\widetilde{\mathbf{z}}) \quad = \quad p_{\boldsymbol{\zeta}}(\mathbf{x} - \mathbf{f}(\widetilde{\mathbf{z}})) \quad (8)
\end{array}
$$

$$
p_{\mathbf{T}, \boldsymbol{\lambda}}(\widetilde{\mathbf{z}}|\mathbf{u}) = \begin{cases} \frac{Q(\widetilde{\mathbf{z}}^{\leq m})e^{<\mathbf{T}(\widetilde{\mathbf{z}}^{\leq m}), \boldsymbol{\lambda}(\mathbf{u})>}}{Z(\mathbf{u})} \\ \mathcal{N}(\mathbf{0}_{(d-m)\times 1}, \mathbf{I}_{(d-m)\times(d-m)}) \end{cases} \quad (9)
$$

where $\boldsymbol{\theta} = (\mathbf{f}, \mathbf{T}, \boldsymbol{\lambda}) \in \boldsymbol{\Theta}$ are model parameters. Equation (7) describes the process of generating $\mathbf{x}$ from $\mathbf{z}$. Equation (8) indicates that $\mathbf{x} = \mathbf{f}(\widetilde{\mathbf{z}}) + \boldsymbol{\zeta}$, where $p_{\boldsymbol{\zeta}}(\boldsymbol{\zeta}) = \mathcal{N}(\mathbf{0}, \mathbf{I})$ and the decoder $\mathbf{f}(\widetilde{\mathbf{z}})$ is assumed to be an invertible function, which is approximated by a neural network. As presented in equation (9), we use the exponential conditional distribution (Pacchiardi & Dutta, 2022) for the first $m$ dimensions and another distribution (e.g., standard normal distribution) for the remaining $d - m$ dimensions to capture other non-interest factors for generation , where $\mathbf{T} : \mathbb{R}^d \to \mathbb{R}^{d \times l}$ is the sufficient statistic, $\boldsymbol{\lambda} : \mathbb{R}^k \to \mathbb{R}^{d \times l}$ is the corresponding parameter, $Q : \mathbb{R}^d \to \mathbb{R}$ is the base measure, $Z(\mathbf{u})$ is the normalizing constant and $< \cdot, \cdot >$ denotes the dot product. If $d = m$, we will only use

the conditional prior in the first line of (9). When causal relationships exist among the generative factors of data $\mathbf{x}$, indicating their non-mutual independence, incorporating information $\mathbf{u}$ alters the prior distribution from a factorial distribution to better match the real-world situation.

We define the inference model that utilizes causal flows as follows:

$$q_{\boldsymbol{\phi}}(\mathbf{z}|\mathbf{x}, \mathbf{u}) \quad = \quad q_{\boldsymbol{\epsilon}}(\mathbf{z} - \boldsymbol{\phi}(\mathbf{x}, \mathbf{u})) \tag{10}$$

$$\mathbf{z} \quad \sim \quad q_{\boldsymbol{\phi}}(\mathbf{z}|\mathbf{x}, \mathbf{u}) \tag{11}$$

$$\widetilde{\mathbf{z}} \quad = \quad \boldsymbol{g}(\mathbf{z}, \mathbf{x}) \tag{12}$$

$$q_{\boldsymbol{\phi}, \boldsymbol{\gamma}}(\widetilde{\mathbf{z}}|\mathbf{x}, \mathbf{u}) \quad = \quad q_{\boldsymbol{\phi}}(\mathbf{z}|\mathbf{x}, \mathbf{u}) \prod_{i=1}^{d} \exp(-s_i(\widetilde{\mathbf{z}} \circ A_{i,:}, \mathbf{x})) \tag{13}$$

where we use $\boldsymbol{\gamma} = (\boldsymbol{s}, \boldsymbol{t}, A) \in \boldsymbol{\Gamma}$ to denote parameters of causal flows. Equation (10) indicates that $\mathbf{z} = \boldsymbol{\phi}(\mathbf{x}, \mathbf{u}) + \boldsymbol{\epsilon}$, where the probability density of $\boldsymbol{\epsilon}$ is $q_{\boldsymbol{\epsilon}}(\boldsymbol{\epsilon}) = \mathcal{N}(\mathbf{0}, \mathbf{I})$ and $\boldsymbol{\phi}(\mathbf{x}, \mathbf{u})$ denotes the encoder. Equation (11) and (12) describe the process of transforming the original encoder output $\mathbf{z}$ into the final latent variable representation $\widetilde{\mathbf{z}}$ by using causal flows. Eventually, the posterior distribution obtained by the inference model is represented by equation (13). Now, the parameters of stochastic transformation $E(\mathbf{x}, \mathbf{u})$ are $\boldsymbol{\phi}$ and $\boldsymbol{\gamma}$.

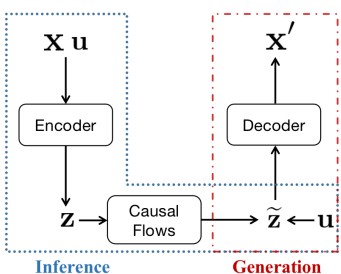

Now we suppose that the dataset $\mathcal{X}$ has an empirical data distribution denoted by $q_{\mathcal{X}}(\mathbf{x}, \mathbf{u})$. Our goal turns into maximizing the variational lower bound on the marginal likelihood $p_{\boldsymbol{\theta}}(\mathbf{x}|\mathbf{u})$. When we have labels $\boldsymbol{y}$, we can add a regularization term in the objective function to promote the consistency between $\boldsymbol{\xi}$ and $E(\mathbf{x})$. Therefore, the loss function of **CauF-VAE** is formulated as follows:

Figure 1: Model structure of CauF-VAE.

$$\begin{aligned} \mathcal{L}(\boldsymbol{\phi}, \boldsymbol{\gamma}, \boldsymbol{\theta}) \quad &= \quad -\text{ELBO}(\boldsymbol{\phi}, \boldsymbol{\gamma}, \boldsymbol{\theta}) + \beta_{sup}\mathcal{L}_{sup}(\boldsymbol{\phi}, \boldsymbol{\gamma}) \\ &= \quad -\mathbb{E}_{q_{\mathcal{X}}} \left[ \mathbb{E}_{q_{\boldsymbol{\phi}, \boldsymbol{\gamma}}(\widetilde{\mathbf{z}}|\mathbf{x}, \mathbf{u})} \log p_{\mathbf{f}}(\mathbf{x}|\widetilde{\mathbf{z}}, \mathbf{u}) - D_{\text{KL}}(q_{\boldsymbol{\phi}, \boldsymbol{\gamma}}(\widetilde{\mathbf{z}}|\mathbf{x}, \mathbf{u}) \| p_{\mathbf{T}, \boldsymbol{\lambda}}(\widetilde{\mathbf{z}}|\mathbf{u})) \right] \\ &+ \quad \beta_{sup}\mathbb{E}_{(\mathbf{x}, \boldsymbol{y}, \mathbf{u})} \left[ l_{sup}(\boldsymbol{\phi}, \boldsymbol{\gamma}) \right] \end{aligned} \tag{14}$$

where $\beta_{sup} > 0$ is a hyperparameter, $l_{sup}(\boldsymbol{\phi}, \boldsymbol{\gamma}) = \sum_{i=1}^{m} -\boldsymbol{y}^i \log \sigma(\bar{E}(\mathbf{x}, \mathbf{u})^i) - (1 - \boldsymbol{y}^i) \log (1 - \sigma(\bar{E}(\mathbf{x}, \mathbf{u})^i))$ is the cross-entropy loss if $\boldsymbol{y}^i$ is the binary label, and $l_{sup}(\boldsymbol{\phi}, \boldsymbol{\gamma}) = \sum_{i=1}^{m} (\boldsymbol{y}^i - \bar{E}(\mathbf{x}, \mathbf{u})^i)^2$ is the Mean Squared Error (MSE) if $\boldsymbol{y}^i$ is the continuous observation. The loss term $\mathcal{L}_{sup}$ aligns the factor of interest $\boldsymbol{\xi} \in \mathbb{R}^m$ with the first $m$ dimensions of the latent variable $\mathbf{z}$, in order to satisfy the Definition 1 (Locatello et al., 2020; Shen et al., 2022).

## 4.2 DISENTANGLEMENT IDENTIFIABILITY

We establish the identifiability of disentanglement of CauF-VAE, which confirms that our model can learn disentangled representations. As we only focus on disentangling the factors of interest, we will, for simplicity, present our proposition in the case where $d = m$. Here, we consider the deterministic part of the posterior distribution as the learned representation.

**Proposition 1** *Under the assumptions of infinite capacity for $E$ and $\mathbf{f}$, the solution $(\boldsymbol{\phi}^*, \boldsymbol{\gamma}^*, \boldsymbol{\theta}^*) \in \text{argmin}_{\boldsymbol{\phi}, \boldsymbol{\gamma}, \boldsymbol{\theta}} \mathcal{L}(\boldsymbol{\phi}, \boldsymbol{\gamma}, \boldsymbol{\theta})$ of the loss function (14) guarantees that $\bar{E}_{\boldsymbol{\phi}^*, \boldsymbol{\gamma}^*}(\mathbf{x})$ is disentangled with respect to $\xi$, as defined in Definition 1.*

*Proof* See Appendix A.2.

It's important to note that our model utilizes two types of supervised information: extra information $\mathbf{u}$ and the labels $\boldsymbol{y}$ of the ground-truth factors. In our experiments, we set $\mathbf{u}$ to be the same as $\boldsymbol{y}$, so that we only need to use one type of supervised information. With the help of the supervised information, we achieve disentanglement identifiability by using the conditional prior and additional regularization terms, which further guarantees that $\widetilde{\mathbf{z}}^i$ is a causal disentangled representation, for $1 \le i \le m$. This means that the latent variables can capture the true underlying factors successfully.

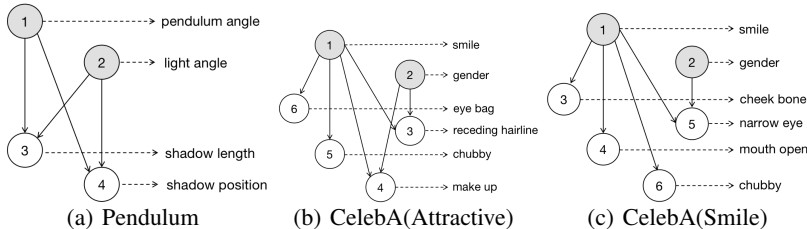

Figure 2: Causal graphs of Pendulum and CelebA. The gray circles represent the causal variables in the graphs. In Figures (a), (b), and (c), we label the underlying factors we are interested in each dataset.

## 5 RELATED WORK

**Causally disentangled representation learning based on VAE**   Recently, achieving causally disentangled representations by VAE has received wide attention. However, the exploration of related methods remains relatively limited. The disentangled causal mechanisms investigated in Suter et al. (2019) and Reddy et al. (2022) assumed that the underlying factors were conditionally independent given a shared confounder. Our proposed model, in contrast, considers more general scenarios where the generative factors can exhibit more complex causal relationships. CausalVAE (Yang et al., 2021) designed a SCM layer to model the causally generative mechanisms of data. DEAR (Shen et al., 2022) used a SCM to construct prior distribution, and employs Generative Adversarial Networks (GAN) to train the model. Unlike SCM-based methods, we leverage the intrinsic properties of flow models to achieve disentanglement and do not rely on external algorithms for training. Later, it has been proposed that a disentangled decoder needs to be trained (An et al., 2023). However, our main focus is on representation learning, which means we give greater importance to the disentangled representations generated by the encoder.

**Causal structure in flows**   We could efficiently learn the causal direction between two variables by using autoregressive flows. Motivated by this, we design causal flows. In a separate parallel work, Wehenkel & Louppe (2021) proposed a generalized graphical normalizing flows. They also utilized a conditioner that incorporated an adjacency matrix. However, we focus on developing the causal conditioner to incorporate causal structure knowledge into flows for achieving causally disentangled representation in VAE. Instead, they only explored the relationship between different flows from Bayesian network perspective and designed a graph conditioner primarily for better density estimation.

## 6 EXPERIMENTS

We empirically evaluate CauF-VAE, and demonstrate that the learned representations are causally disentangled, enabling the model to perform well on various tasks. Our experiments are conducted on both synthetic dataset and real human face image dataset, and we compare CauF-VAE with some state-of-the-art VAE-based disentanglement methods.

### 6.1 EXPERIMENTAL SETUP

We utilize the same datasets as Shen et al. (2022) where the underlying generative factors are causally related. The synthetic dataset is Pendulum, with four continuous factors whose causal graph of the factors is shown in Figure 2(a). The training and testing sets consist of 5847 and 1461 samples, respectively. The real human face dataset is CelebA (Liu et al., 2015), with 40 discrete labels. We consider two sets of causally related factors named CelebA(Attractive) and CelebA(Smile) with causal graphs also depicted in Figure 2(b) and 2(c). The training and testing sets consist of 162770 and 19962 samples, respectively.

We compare our method with several representative VAE-based models for disentanglement (Locatello et al., 2019a), including $\beta$-VAE (Higgins et al., 2017), $\beta$-TCVAE (Chen et al., 2018), and DEAR (Shen et al., 2022). We also compare with vanilla VAE (Kingma & Welling, 2013). To ensure a fair

comparison with equal amounts of supervised information, for each of these methods we use the same conditional prior and loss term as in CauF-VAE. For comprehensive implementation details and hyperparameters, please refer to Appendix B.

## 6.2 EXPERIMENTAL RESULTS

Now, we proceed to evaluate our method from several aspects and provide an analysis of the corresponding experimental results.

### 6.2.1 CAUSALLY DISENTANGLED REPRESENTATIONS

To qualitatively verify that CauF-VAE indeed learns causally disentangled representations, we conduct intervention experiments. Intervention experiments involve performing the "do-operation" in causal inference (Pearl, 2009b). Taking a single-step causal flow as an example, we demonstrate step by step how our model performs "do-operation". First, given a trained model, we input the sample $\mathbf{x}$ into the encoder, obtaining an output $\mathbf{z}$. Assuming we wish to perform the "do-operation" on $\widetilde{\mathbf{z}}^i$, i.e., $do(\widetilde{\mathbf{z}}^i = c)$, we follow the approach in Khemakhem et al. (2021) by treating equation (4) as SEMs. Specifically, we set the input and output of $\widetilde{\mathbf{z}}^i$ to the control value $c$, while other values are computed iteratively from input to output. Finally, the resulting $\widetilde{\mathbf{z}}$ is decoded to generate the desired image, which corresponds to generating images from the interventional distribution of factor $\widetilde{\mathbf{z}}$.

We perform intervention experiments by applying the "do-operation" to $m-1$ variables in the first $m$ dimensions of the latent variables, resulting in the change of only one variable. This operation, which has been referred to as "traverse", aims to test the disentanglement of our model (Shen et al., 2022). Figures 3 and 4 show the experimental results of the CauF-VAE and DEAR on Pendulum and CelebA(Smile). We observe that when traversing a latent variable dimension, CauF-VAE has almost only one factor changing, while DEAR has multiple factors changing. This is clearly shown by comparing "traverse" results of the third row for shadow length in Figures 3(a) and 3(b), as well as the second row for gender in Figures 4(a) and 4(b). Therefore, our model can better achieve causal disentanglement.

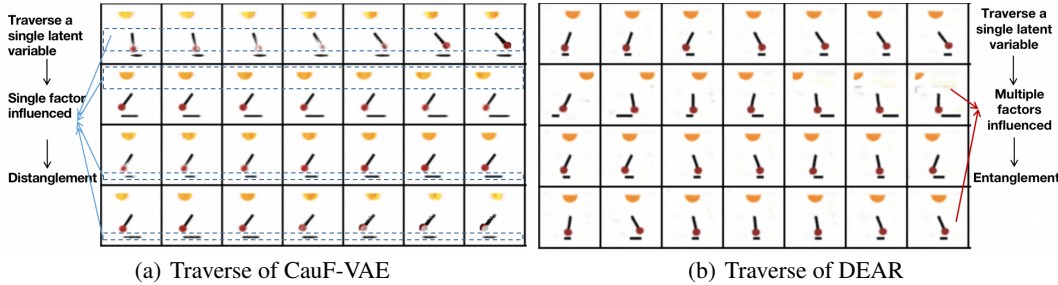

(a) Traverse of CauF-VAE            (b) Traverse of DEAR

Figure 3: Results of traverse experiments on Pendulum. Each row corresponds to a variable that we traverse on, specifically, pendulum angle, light angle, shadow length, and shadow position.

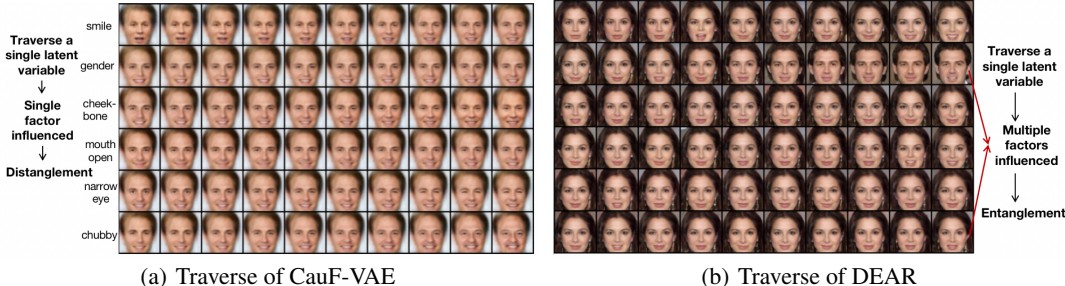

(a) Traverse of CauF-VAE            (b) Traverse of DEAR

Figure 4: Results of traverse experiments on CelebA(Smile). Each row corresponds to a variable that we traverse on, specifically, smile, gender, cheek bone, mouth open, narrow eye and chubby.

To demonstrate our model's capability to perform interventions hence generating new images beyond the dataset, we conduct "do-operations" on individual latent variables. Figure 5 illustrates these operations, with each row representing an intervention on a single dimension. In Figure 5(a), we observe that intervening on the pendulum angle and light angle produces changes in shadow length in accordance with physical principles. However, intervening on shadow length has minimal impact on these two factors. Similarly, as seen in Figure 5(b), intervening on gender influences narrow eye appearance, but the reverse is not true. This demonstrates that intervening on causal factors affects the resulting effects, but not the other way around. Hence, our latent variables have effectively learned factor representations, attributed to the design of the causal flows, which incorporate $A$. Additional traversal and intervention results are presented in Appendix C.

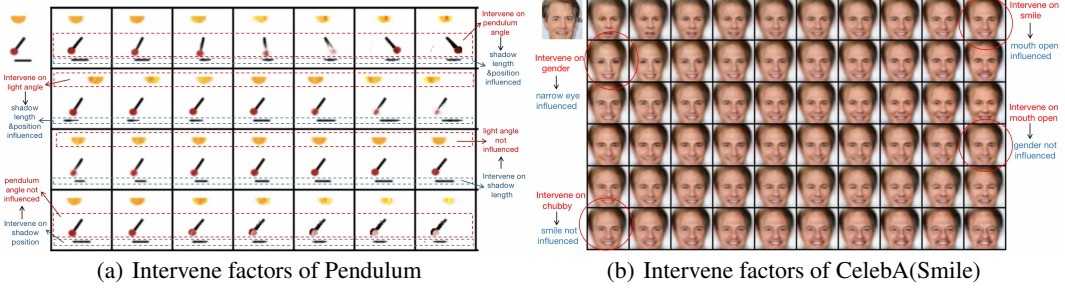

(a) Intervene factors of Pendulum  (b) Intervene factors of CelebA(Smile)

Figure 5: Results of intervention on only one variable for both Pendulum and CelebA(Smile). The image in the upper left corner of (a) and (b) are the test data we consider respectively.

### 6.2.2 DOWNSTREAM TASKS

To further illustrate benefits of causally disentangled representations, we consider its impact on downstream tasks in terms of sample efficiency and distributional robustness. We introduce two downstream prediction tasks to compare our model with baseline models. First, for Pendulum, we normalize factors to $[-1, 1]$ during preprocessing. Then, we manually create a classification task: if $pendulum\ angle > 0$ and $light\ angle > 0$, the target label $y = 1$; otherwise, $y = 0$. For the CelebA(Attractive), we adopt the same classification task as in Shen et al. (2022). We employ multilayer perceptron (MLP) to train classification models, where both the training and testing sets consist of the latent variable representation $\widetilde{\mathbf{z}}$ and their corresponding labels $\mathbf{y}$.

**Sample Efficiency** We adopt the statistical efficiency score defined in Locatello et al. (2019a) as a measure of sample efficiency, which is defined as the classification accuracy of 100 test samples divided by the number of all (Pendulum)/10,000 test samples (CelebA). The experimental results are presented in Table 1. Table 1 shows that CauF-VAE achieves the best sample efficiency and test accuracy on both datasets, except for the test accuracy of all test samples of Pendulum where $\beta$-VAE is the best. However, on the complex CelebA dataset, CauF-VAE significantly outperforms $\beta$-VAE. We attribute the superiority of our model to our modeling approach, i.e., leveraging the fitting ability of flows, especially causal flows that greatly enhance the encoder's ability to learn semantically meaningful representations.

Table 1: Test accuracy and sample efficiency of different models on Pendulum and CelebA datasets. Mean±standard deviations are included in the Table.

| | **Pendulum** | | | **CelebA** | | |
|---|---|---|---|---|---|---|
| **Model** | 100(%) | All(%) | Sample Eff | 100(%) | 10000(%) | Sample Eff |
| CauF-VAE | $\mathbf{99.00_{\pm 0}}$ | $99.43_{\pm 0.34}$ | $\mathbf{99.57_{\pm 0.34}}$ | $\mathbf{81.00_{\pm 1.73}}$ | $\mathbf{81.54_{\pm 1.76}}$ | $\mathbf{99.34_{\pm 0.27}}$ |
| DEAR | $88.00_{\pm 0}$ | $88.55_{\pm 0.04}$ | $98.63_{\pm 1.33}$ | $61.00_{\pm 3.60}$ | $68.50_{\pm 0}$ | $89.05_{\pm 5.26}$ |
| $\beta$-VAE | $98.67_{\pm 1.15}$ | $\mathbf{99.59_{\pm 0.07}}$ | $98.94_{\pm 0.92}$ | $62.33_{\pm 5.69}$ | $68.49_{\pm 0.02}$ | $91.01_{\pm 8.28}$ |
| $\beta$-TCVAE | $97.67_{\pm 1.15}$ | $99.38_{\pm 0.48}$ | $98.27_{\pm 0.79}$ | $75.33_{\pm 3.21}$ | $78.72_{\pm 4.93}$ | $95.83_{\pm 4.10}$ |
| VAE | $98.33_{\pm 0.58}$ | $99.48_{\pm 0.39}$ | $98.72_{\pm 0.37}$ | $60.33_{\pm 2.89}$ | $68.50_{\pm 0}$ | $88.08_{\pm 4.21}$ |

**Distributinal robustness** To assess distributional robustness, we modify the controllable synthetic Pendulum dataset during training to inject spurious correlations between the target label and some spurious attributes. We choose $background\_color \in \{blue(+), white(-)\}$ as a spurious feature. Specifically, the target label and the spurious attribute of 80% of the examples are both positive or negative, while those of 20% examples are opposite. For instance, in the manipulated training set, 80% positive examples in Pendulum are masked with a blue background. But in the test set, we do not inject this correlation, rusulting in a distribution shift. The results are summarized in Table 2.

The results include average and worst-case test accuracy, evaluating overall classification performance and distributional robustness. Worst-case accuracy identifies the group with the lowest accuracy among four groups categorized based on target and spurious binary labels. It often involves opposing spurious correlations compared to training data. The classifiers trained using CauF-VAE representations demonstrate significant superiority over the baseline models in both evaluation metrics. Notably, CauF-VAE experiences a smaller decrease in worst-case accuracy compared to average accuracy, indicating robustness to distributional shifts.

Table 2: Distributional robustness of different models.

| Model | TestAvg(%) | TestWorst(%) |
|---|---|---|
| CauF-VAE | $\mathbf{97.83_{\pm 1.18}}$ | $\mathbf{94.70_{\pm 3.41}}$ |
| DEAR | $80.40_{\pm 0.47}$ | $64.50_{\pm 2.67}$ |
| $\beta$-VAE | $96.48_{\pm 2.06}$ | $90.05_{\pm 5.44}$ |
| $\beta$-TCVAE | $96.57_{\pm 1.33}$ | $90.27_{\pm 3.74}$ |
| VAE | $95.14_{\pm 3.46}$ | $88.81_{\pm 5.44}$ |

### 6.2.3 EXPLORING THE POTENTIAL FOR LEARNING THE STRUCTURE OF A

Apart from the aforementioned applications, CauF-VAE has the potential to learn true causal relationships between factors. As shown in Figure 7(a)-7(d), for Pendulum, when our model $A$ adopts the super-graph shown in Figure 6, though the corresponding $A$ is initialized randomly around 0, it gradually approaches the true causal structure during the training process. If we use a threshold of 0.2 and prune edges in the causal graph with values below this threshold, we obtain Figure 7(e), which corresponds to the true causal structure. For details, see Appendix C.

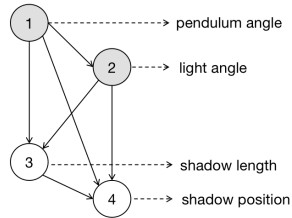

Figure 6: Super-graph of Pendulum.

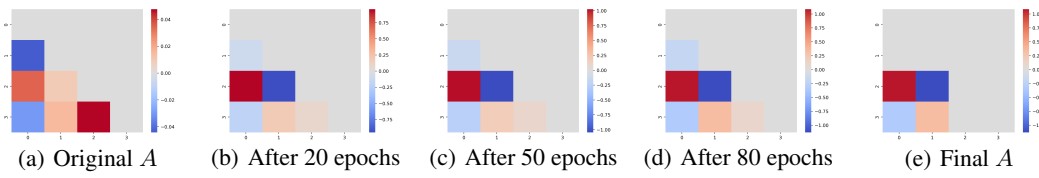

| (a) Original $A$ | (b) After 20 epochs | (c) After 50 epochs | (d) After 80 epochs | (e) Final $A$ |

Figure 7: The learned weighted adjacency matrix $A$ given a super-graph on Pendulum. (a)-(d) illustrate the changes in $A$ as the training progresses. (e) represents $A$ after edge pruning.

## 7 CONCLUSION

This paper focuses on learning causally disentangled representations where the underlying generative factors are causally related. We introduce causal flows that incorporate causal structure information of factors, and propose CauF-VAE, a powerful model for learning causally disentangled representations. By adding additional supervised information, our model theoretically achieves disentanglement identifiability. To the best of our knowledge, our method is the first to achieve causal disentanglement without relying on complex Structural Causal Models (SCMs), while also not limiting the causal graph of factors. This suggests an interesting direction for future work. A potential step is to deeply explore the data to learn low-dimensional representations of causal factors. Additionally, designing unsupervised models that can do this can also be considered as future reasearch.

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
