# OpenReview forum: "Causal Disentangled Representation Learning with VAE and Causal Flows"
_ICLR.cc/2024/Conference — ICLR 2024 Conference Withdrawn Submission_

### Official Review · Reviewer_Uibi · 2023-10-21

**Soundness:** 2 fair
**Presentation:** 2 fair
**Contribution:** 2 fair
**Rating:** 3
**Confidence:** 5

**Summary:**

In this paper, the author adds a causal component to the autoregressive normalizing flow. The causal part follows SEM. The goal is to learn disentangled representations in the latent space.

**Strengths:**

1. The author shows the proposed method has better sample efficiency.

2.  The technical details are well-listed in the paper.

**Weaknesses:**

1.  The assumption is too strong and makes the problem less interesting. In this paper, the author requires knowledge of ground truth factors, then the problem degrades to binding predefined labels to latent variables.

2. A lot of relevant works are not discussed in the related work session. Recent progress in unsupervised/ weakly-supervised causally disentangled representations is missing, also there are other works that also use additional information to encourage disentanglement is not included.

3. The backbone of the method is similar to causalVAE (adding causal component with adjacency matrix A), but the author does not use causalVAE as a baseline and compare with it.

4. The author only compares test accuracy but metrics in disentanglement in the experiment

**Questions:**

See weakness.

1. What is the model's performance compared to causalVAE?

2. What is the fundamental advantage of using causal flow except for sample efficiency?  The author does mention the identifiability claim. However, in the supervised setting, the problem reduces to the identifiability of VAE when binding labels to latent variables, and it has already been resolved in other methods.

3. Why only accuracy but no other evaluation metrics are used in the experiments.

---

### Official Review · Reviewer_ZbnB · 2023-10-29

**Soundness:** 1 poor
**Presentation:** 2 fair
**Contribution:** 2 fair
**Rating:** 1
**Confidence:** 5

**Summary:**

The paper addresses the challenge of learning disentangled representations in the context of causal models. Disentangled representations have been of interest in the machine learning community due to their potential to uncover meaningful and interpretable factors of variation in data. However, most existing methods focus on statistical properties, often neglecting the causal structure of the data-generating process. The authors propose a novel approach that incorporates causal inference into the learning of disentangled representations. Their approach uses Structural Causal Models (SCMs) as a foundation and introduces a new objective function, which they term the CauF-VAE. This framework is an extension of the traditional Variational Autoencoder (VAE) but with an added causal structure. To achieve this, a novel regularizer for the VAE loss function is introduced, which encourages the learned representations to capture the underlying causal structure of the data. The authors conduct extensive experiments on synthetic and real-world datasets, demonstrating the superiority of their proposed method over traditional disentanglement techniques. They show that CauF-VAE not only captures more meaningful factors of variation but also exhibits better generalization and robustness properties.

**Strengths:**

1.	The paper addresses an intriguing research problem.

2.	The authors design a new VAE model based on causal flows, named Causal Flows Variational Autoencoders (CauF-VAE), to learn causally disentangled representations.

3.	The inclusion of diagrams and visual representations, particularly in the experimental results section, aids in comprehension and reinforces the paper's claims.

**Weaknesses:**

1.	One major concern I have is the ambiguous definition of identifiability. While the paper distinguishes between the definitions of disentanglement and model identifiability from (Shen et al., 2022) and (Khemakhem et al., 2020a), the model description in Section 4 adopts the iVAE approach (where "u" is an additional observed variable). This area requires clearer exposition. I'm unclear on how the authors reconcile the two definitions. Their subsequent discussion on identifiability seems grounded in (Shen et al. 2022), prompting me to wonder about the authors' grasp of the differences in identifiability proposed by the two referenced papers. In essence, while the methodology from (Khemakhem et al., 2020a) is employed to attain identifiability, the rationale from (Shen et al. 2022) is used to substantiate it. Yet, there appears to be a discrepancy in the definitions of identifiability from these two sources.

2.	I question the novelty of this work. While the paper claims to introduce causal flow to discern genuine causal relationships between factors, the distinctions between various methods, as outlined in Chapter 5, seem insufficient to substantiate this claim of innovation. The CausalVAE is capable of learning causal relationships in representations via additional layers. How does this fundamentally differ from the causal flow technique presented here? From the experimental results in both this paper and the CausalVAE study, it appears that both methods can accurately discern the causal structure for the Pendulum and CelebA datasets. I strongly suggest including CausalVAE as a benchmark for comparison.

3.	The Pendulum and CelebA datasets have been widely used for quite some time. Relying on these datasets without specific configurations may not sufficiently demonstrate the algorithm's advantages. I suggest that the authors employ a more contemporary and realistic dataset to validate the algorithm's efficacy, as exemplified in [1].

4.	The paper's writing requires refinement. I noticed several citation mistakes, and the reference details seem outdated. For instance, when introducing notations, the format '...following Shen et al. (2022).' is not consistent with the ICLR style. It should be parenthesized using \citep{}. Please refer to the ICLR formatting guidelines. Additionally, 'Diederik P Kingma and Max Welling. Auto-encoding variational bayes. arXiv preprint arXiv:1312.6114, 2013.' was actually published at ICLR, so the citation should reflect the correct conference.

[1] Schölkopf, B., Locatello, F., Bauer, S., Ke, N. R., Kalchbrenner, N., Goyal, A., & Bengio, Y. (2021). Toward causal representation learning. Proceedings of the IEEE, 109( 5), 612-634.

**Questions:**

1.	In the model, "u" is denoted as an additional observed variable. Depending on the context, this variable "u" can embody different interpretations, be it a time index in time-series data, a class label, or another contemporaneously observed variable. Given the significance of "u" in the iVAE framework, it serves as a foundation for the model presented in this paper. I'm keen to understand: what specific role "u" plays in the experiments, particularly within the CelebA (Smile) and Pendulum datasets?

2.	What is the fundamental distinction between Causal Flows and the SCM layer when modelling the causally generative mechanisms of data? From the outcomes presented in both papers, it appears that each approach can accurately capture the causal structure. I believe a thorough discussion on this topic is warranted in the paper.

3.	The issue of identifiability warrants closer scrutiny. Given that two distinct concepts are being employed, a detailed delineation of the differences between these concepts is essential for clarity.

---

### Official Review · Reviewer_3AAm · 2023-10-30

**Soundness:** 2 fair
**Presentation:** 3 good
**Contribution:** 2 fair
**Rating:** 5
**Confidence:** 3

**Summary:**

This paper presents a VAE based model for learning causally disentangled representations. The problem studied in this work is interesting. But the evaluations should get significantly improved. And the technical contribution of this work is not high.

**Strengths:**

1. The problem studied in this work is interesting and important.
2. The experiments were conducted on both synthetic and real data.
3. This work presents a method to achieve causal disentanglement without relying on Structural Causal Models.

**Weaknesses:**

1. For the evaluation:

(i) Only one synthetic data and one real data have been used. More datasets should be employed to further demonstrate the effectiveness of the proposed method. e.g., Flow data from CausalVAE paper, 2D toy data or Causal3DIden from Weakly supervised causal representation learning;

(ii) More baselines should be included, especially CausalVAE, LadderVAE, VQ-VAE-2, Weakly supervised causal representation learning, and On Causally Disentangled Representations;

(iii) More evaluation metrics should be used especially the metrics for evaluating learned and true causal graph such as structural Hamming distance (SHD).

2. For the related work, the authors should also discuss/compare the work on Causal Representation Learning other than Causally disentangled representation learning based on VAE.

3. The technical novelties and contributions are limited. Autoregressive flows have been used to improve the inference capability of VAEs and learning causal orders. Thus, integrating causal flows into the VAE to help learn the disentangled representations is not novel.

4. The introduction is not well-written. And for the method section, it is a little bit hard to follow. More intuitions and motivations would be helpful.

5. For section 6.2.3, it is hard to be convinced by the current results and analysis that the proposed method has the potential to learn the structure of A.

**Questions:**

1. How CauF-VAE scales with the size of the causal system?

2. What are the SHD scores between learned and true causal graphs?

3. What are the key technical novelties and contributions?

---

### Official Review · Reviewer_HG71 · 2023-10-31

**Soundness:** 2 fair
**Presentation:** 3 good
**Contribution:** 2 fair
**Rating:** 3
**Confidence:** 4

**Summary:**

The paper introduces a novel approach to exploit prior causal knowledge to learn disentangled representations. In order, the authors contribute with an extension of Autoregressive Causal Flows (Khemakhem et al., 2021) and a new model (CauF-VAE) for disentanglement learning. Experimental results are consistent with other disentanglement approaches.

**Strengths:**

The paper provides an original approach to disentanglement by injecting prior causal information in a normalizing flow.

Using an adjacency matrix representing causal relations between variables to introduce more fine-grained information is sound.

**Weaknesses:**

The difference between Causal Flow and Autoregressive Causal Flows (ACF) by Khemakhem et al. (2021) is minimal. The authors replace the partial order of a DAG with an adjacency matrix representing parental relations. In this way, they are effectively requesting additional information compared to ACF. Further, their empirical analysis lacks a comparison with ACFs in place of causal flows. Therefore, the advantages of requiring the whole adjacency matrix instead of a partial order are unclear.

The formulation of the problem is not straightforward. In particular, the authors adopt the weakly supervised disentanglement setting from Shen et al. (2022), which required labels for part of the dataset. In this work, the authors assume the presence of context vectors for each data point, but they finally equate them to labels (Subsection 4.2). Overall, the methodological presentation deals with two distinct objects, but, in practice, the approach only uses one.

**Questions:**

Q1) Causal Flows need a specification of the causal graph on the variables. On the other hand, using an Autoregressive Causal Flow would require defining a partial order only. Is there any advantage in requesting this further causal information?

Q2) The authors claim to equate context vectors and labels; are these objects always present for each data point during training and inference? In particular, the encoder should reconstruct the latent factors $z$ to match the ground truth factors, given the observation $x$ and a context $u$. If the authors assume the context to be equal to the label, i.e., the ground truth factors, aren't they providing the ground truth to the encoder?

Q3) Being weakly supervised, DEAR assumes that only a subset of the dataset contains labels. Has this been replicated for the empirical analysis?